# Challenges Addressing Inequalities in Measles Vaccine Coverage in Zambia through a Measles–Rubella Supplementary Immunization Activity during the COVID-19 Pandemic

**DOI:** 10.3390/vaccines11030608

**Published:** 2023-03-07

**Authors:** Yangyupei Yang, Natalya Kostandova, Francis Dien Mwansa, Chola Nakazwe, Harriet Namukoko, Constance Sakala, Patricia Bobo, Penelope Kalesha Masumbu, Bertha Nachinga, David Ngula, Andrea C. Carcelen, Christine Prosperi, Amy K. Winter, William J. Moss, Simon Mutembo

**Affiliations:** 1Department of International Health, International Vaccine Access Center, Johns Hopkins Bloomberg School of Public Health, Baltimore, MD 21231, USA; 2Department of Epidemiology, Johns Hopkins Bloomberg School of Public Health, Baltimore, MD 21231, USA; 3Ministry of Health, Government of the Republic of Zambia, Lusaka 10101, Zambia; 4Zambia Statistics Agency, Lusaka 10101, Zambia; 5Zambia Country Office, World Health Organization, Lusaka 10101, Zambia; 6Department of Epidemiology, University of Georgia, Athens, GA 30602, USA; 7W Harry Feinstone Department of Molecular Microbiology and Immunology, Johns Hopkins Bloomberg School of Public Health, Baltimore, MD 21231, USA

**Keywords:** zero-dose, under-immunization, measles, rubella, vaccination, supplementary immunization activity, campaign, children, inequality

## Abstract

Background: Measles–rubella supplementary immunization activities (MR-SIAs) are conducted to address inequalities in coverage and fill population immunity gaps when routine immunization services fail to reach all children with two doses of a measles-containing vaccine (MCV). We used data from a post-campaign coverage survey in Zambia to measure the proportion of measles zero-dose and under-immunized children who were reached by the 2020 MR-SIA and identified reasons associated with persistent inequalities following the MR-SIA. Methods: Children between 9 and 59 months were enrolled in a nationally representative, cross-sectional, multistage stratified cluster survey in October 2021 to estimate vaccination coverage during the November 2020 MR-SIA. Vaccination status was determined by immunization card or through caregivers’ recall. MR-SIA coverage and the proportion of measles zero-dose and under-immunized children reached by MR-SIA were estimated. Log-binomial models were used to assess risk factors for missing the MR-SIA dose. Results: Overall, 4640 children were enrolled in the nationwide coverage survey. Only 68.6% (95% CI: 66.7%, 70.6%) received MCV during the MR-SIA. The MR-SIA provided MCV1 to 4.2% (95% CI: 0.9%, 4.6%) and MCV2 to 6.3% (95% CI: 5.6%, 7.1%) of enrolled children, but 58.1% (95% CI: 59.8%, 62.8%) of children receiving the MR-SIA dose had received at least two prior MCV doses. Furthermore, 27.8% of measles zero-dose children were vaccinated through the MR-SIA. The MR-SIA reduced the proportion of measles zero-dose children from 15.1% (95% CI: 13.6%, 16.7%) to 10.9% (95% CI: 9.7%, 12.3%). Zero-dose and under-immunized children were more likely to miss MR-SIA doses (prevalence ratio (PR): 2.81; 95% CI: 1.80, 4.41 and 2.22; 95% CI: 1.21 and 4.07) compared to fully vaccinated children. Conclusions: The MR-SIA reached more under-immunized children with MCV2 than measles zero-dose children with MCV1. However, improvement is needed to reach the remaining measles zero-dose children after SIA. One possible solution to address the inequalities in vaccination is to transition from nationwide non-selective SIAs to more targeted and selective strategies.

## 1. Introduction

Measles and rubella remain important causes of morbidity and mortality. In 2021, there were an estimated 9.5 million measles cases and 128,000 measles deaths globally [1]. Rubella results in an estimated 100,000 annual deaths globally due to congenital rubella syndrome [2]. Both viruses are endemic in sub-Saharan Africa, and the elimination targets for all member states in the World Health Organization (WHO) Africa region are far from being reached [3].

Zambia has made significant progress in increasing coverage with measles-containing vaccines (MCVs) over the past two decades and attained high routine vaccination coverage of 93% with the first dose of MCV (MCV1) in 2019. However, routine vaccination coverage with the second dose of MCV (MCV2) has lagged behind at approximately 66% [4]. Following widespread disruptions to immunization services because of the COVID-19 pandemic, there are concerns that children who missed their routine and campaign vaccination doses could form clusters of susceptible populations that drive measles outbreaks [5,6,7,8]. To fill these population immunity gaps that arise when routine immunization services fail to reach all children with two doses of MCV, the Zambian Ministry of Health has conducted a nationwide non-selective measles–rubella supplementary immunization activity (MR-SIA) every four years since 2003 to avoid accumulation of a cohort of measles-susceptible children. The most recent MR-SIA was conducted in November 2020 during the COVID-19 pandemic, targeting children between the ages of 9 and 59 months [9].

After the MR-SIA, a nationwide, population-level post-campaign coverage survey (PCCS) was conducted in October 2021 as required by Gavi [10]. A PCCS should typically be conducted within three months of SIA completion to measure MR-SIA coverage, defined as the proportion of children in the target age group who received an MR vaccine dose during the SIA. The PCCS for the 2020 MR-SIA was delayed and conducted almost a year after the SIA due to the COVID-19 pandemic. For a PCCS to measure progress towards the goals of the Immunization Agenda 2030, a global strategy “to leave no one behind”, it is critical to estimate MR-SIA coverage among subpopulations who had not previously received MCV from routine services (referred to as measles zero-dose children) and subpopulations of those eligible for MCV2 who had not received MCV2 before the MR-SIA (measles under-immunized children) [11,12]. We used national, cross-sectional data from the PCCS to understand how MR-SIA can address vaccination inequalities by estimating routine and SIA MCV coverages, as well as the proportion of measles zero-dose and under-immunized children reached by the MR-SIA, and identified reasons associated with missing the MR-SIA. Our study will help program managers and researchers understand vaccination inequalities that are overlooked when implementing SIA and when measuring the impact of SIAs using the PCCS.

## 2. Methods

### 2.1. Study Design and Population

Using data from a nationwide, cross-sectional survey conducted in Zambia, we estimated the proportion of measles zero-dose and under-immunized children who were vaccinated during the 2020 MR-SIA. The PCCS was conducted in October 2021 following a non-selective, nationwide MR-SIA in November 2020. The survey enrolled children between the ages of 9 and 59 months at the time of the November 2020 SIA. Therefore, all children born between December 2015 and the end of February 2020 residing in the selected households were eligible for the survey.

### 2.2. Sampling Strategy

Sample selection was based on a two-stage stratified cluster sample design. In total, 110 and 162 enumeration areas (EAs) were selected within urban and rural strata, respectively, with probability proportional to size. Before selection, EAs were sorted by province, district, constituency, ward, rural/urban status, census supervisory area, and standard enumeration area to assure implicit representation.

In the second stage, an updated listing of the households in the selected EAs was generated to ensure that key information in the selected EAs was updated, as well as the accuracy of the number of residential households and households with at least one child aged 9 to 59 months at the time of the SIA. From each EA, 20 households with eligible children were selected using systematic random sampling, and data were collected on all children in the household who met the eligibility criteria. In EAs with 20 or fewer households with eligible children, all households with eligible children were eligible for enrolment (see Supplementary Methods for more detail).

### 2.3. Data Collection

Prior to the interview, informed consent was obtained from the parents or legal guardians. After obtaining parental permission, survey staff collected sociodemographic characteristics and vaccination history from the parents or caregivers of the child using a standardized, tablet-based questionnaire. Receipt of routine and campaign MCV doses was collected by first asking caregivers to recall the child’s vaccination history; then, this was verified by reviewing the under-5 immunization card or any documents (i.e., journals, piece of paper with vaccination notes) where vaccinations were recorded. Standardized questions previously used in demographic and health surveys were asked to collect information on routine and campaign doses of MCV, routine diphtheria–tetanus–pertussis (DTP) vaccine, and routine Bacille Calmette-Guérin (BCG) vaccine.

### 2.4. Statistical Analysis

The weighted study outcomes were, (1) the percentage of children who received the MR vaccine during the SIA, (2) the percentage of measles zero-dose children who received the MR vaccine during the SIA, and (3) MCV1 and MCV2 coverage before and after the SIA. In addition, the impact of the MR-SIA was estimated by calculating the increase in MCV1 and MCV2 coverage after the SIA. For children whose vaccination cards or records were available, we assessed routine vaccination status before the SIA to estimate the proportion of Gavi-defined zero-dose, under-immunized and fully vaccinated children [13]. Gavi defined “zero-dose children” as children who have not received the first dose of the DTP vaccine. “Under-immunized” children were defined as those who missed a third dose of the DTP vaccine. Together, zero-dose and under-immunized children formed missed communities. To assess MCV coverage, we defined “measles zero-dose” children as children who did not receive MCV1. Unvaccinated was defined as children whose parents or caregivers could not recall their children’s vaccination status. Sensitivity analyses were conducted by (1) treating these children as vaccinated and (2) restricting the analysis to children whose vaccination status was card-confirmed.

Log-binomial regression analysis was performed to assess factors associated with missing MR-SIA vaccination. Participant characteristics for enrolled children (sex, setting, age, and DTP vaccination status) and household characteristics (relationship of head of household with enrolled children, maternal education level, maternal COVID-19 vaccination status, and travel time) were included in the univariable analysis to identify social determinants associated with missing MR-SIA vaccination. The prevalence ratio (PR) and the corresponding 95% confidence intervals (95% CI) were calculated. Age-adjusted log-binomial regression was used to address heterogeneity between age categories. Forward and backward stepwise selection methods were used to select the best-fit model. The final multivariable model included variables with a *p*-value < 0.05 from age-adjusted univariable analyses and variables that were of public health importance. Log-likelihood and Akaike’s information criteria (AIC) were used to determine the goodness of fit, and the model with the lowest AIC was selected as the best-fit model. All statistical analyses were performed in R version 4.1.3, and the model was run using the survey package to account for sampling weights [14,15]. Figures were generated using the ggplot2 package [16].

## 3. Results

### 3.1. Characteristics of Survey Respondents

Overall, 5440 households with at least one child younger than 15 years of age at the time of the SIA were selected. Respondents from 5155 (95%) households were available during the survey period. Of the available households, only 4641 (90%) were eligible for the survey after excluding 514 households that had no child between the ages 9 and 59 months. Of these households, only 51 (1%) respondents refused to participate in the survey. After applying the exclusion and inclusion criteria, data were collected on 4640 eligible children during the MR-SIA (Figure 1). After weighting, 49.8% (95% CI: 48.1%, 51.3%) of the children were male, and 59.2% (95% CI: 57.6%, 60.7%) lived in rural areas. About two-fifths (40.5%, 95% CI: 37.6%, 43.1%) did not have vaccination cards or other documents that showed their vaccination status. Overall, 88.2% (95% CI: 87.2%, 89.2%) were fully vaccinated with all recommended DTP doses for their age (Table 1). A proportion of mothers had a primary school education (41.0%, 95% CI: 39.5%, 42.6%), and only 7.7% (95% CI: 6.8%, 8.6%) had an education level higher than secondary school. More than three-quarters of adults (79.1%; 95% CI: 78.2%, 80.0%) who lived in the selected households were not vaccinated against SARS-CoV-2 at the time of PCCS. Only 24.9% (95% CI: 23.5%, 26.3%) of the parents or guardians traveled fewer than 15 min to get to the nearest health facility, regardless of the mode of transport.

### 3.2. SIA Coverage and Added Value in Addressing Inequalities

Based on verbal recall and card-confirmed weighted results, 84.9% (95% CI: 81.8%, 88.0%) of children received at least one dose of routine MCV before the 2020 MR-SIA (Figure 2, Appendix A). Routine MCV1 coverage before the SIA for children between the ages of 9 and 17 months was 83.9% (95% CI: 76.3%, 91.6%) and 85.7% (95% CI: 81.7%, 86.6%) for those 18 months and older (Appendix A). When treating children with unknown MCV vaccination status as vaccinated, the percentage of children who received at least one MCV dose increased from 84.9% to 90.6% (Appendix A).

Overall, 68.6% (95% CI: 66.7%, 70.6%) of children eligible for the 2020 MR-SIA received MCV during the SIA. Only 2.7% of these eligible children had documented evidence of receiving the MR-SIA dose, and the rest were identified by caregiver recall. When routine and SIA MCV coverage was combined, coverage of at least one dose of MCV increased by 4.2%, from 84.9% (95% CI: 81.8%, 88.0%) to 89.1% (95% CI: 86.0%, 92.2%), as assessed by card and verbal recall. Before the MR-SIA, 15.1% (95% CI: 13.6%, 16.8%) of children were measles zero-dose. Of these measles zero-dose children, 27.8% were vaccinated during the SIA when unknown vaccination status was categorized as unvaccinated. This implies that the MR-SIA reduced the proportion of measles zero-dose children from 15.1% (95% CI: 13.6%, 16.7%) to 10.9% (95% CI: 9.7% to 12.3%) (Figure 2, Appendix A). Although the study was not powered for comparisons between provinces, Central Province had the lowest SIA coverage and Eastern and Copperbelt Provinces had the highest SIA coverage (Figure 3A). Western Province had a higher proportion of measles zero-dose children who remained unvaccinated after the SIA (18.9%, 95% CI: 12.8%, 22.5%) (Figure 3B). Prior to the SIA, 59 (8.3%) measles zero-dose children were also DTP zero-dose. After the SIA, 55 (95%) of those children remained zero-dose for both MCV and DTP.

During the SIA, 6.3% (95% CI: 5.6%, 7.1%) of enrolled children received MCV2, and 58.1% (95% CI: 59.8%, 62.8%) received a third or further MCV dose (Figure 4, Appendix A). The sensitivity analysis restricted to children who had a vaccination card or other documentation of vaccines (N = 2762) showed that 1.5% (95% CI: 1.0%, 2.0%) of the children received MCV1 through the SIA, 2.7% (95% CI: 2.1%, 3.4%) of children received MCV2, and 3.6% (95% CI: 2.9%, 4.4%) remained measles zero-dose after the SIA (Appendix A).

### 3.3. Persistent Inequalities after the SIA

Among children eligible for the 2020 MR-SIA, 1454 (31%) were missed by the 2020 MR-SIA (Table 2). Compared to fully vaccinated children using the Gavi definition, zero-dose children were 2.8 times more likely to miss the SIA dose (PR:2.8; 95% CI: 1.8, 4.4), and under-immunized children were 2.2 times more likely to miss the SIA dose (PR:2.2; 95% CI: 1.2, 4.1) after adjusting for age, sex, maternal education level, and travel time to health facilities (Table 2). Children in urban areas were 39% more likely to miss the SIA dose than those in rural areas (PR: 1.39; 95% CI: 1.18, 1.63). Children whose mothers had primary and secondary education were less likely to miss the SIA vaccination than children whose mothers had no formal education (PR: 0.73; 95% CI: 0.58 and 0.92, and PR 0.60; 95% CI: 0.47 and 0.77, respectively) (Table 2). Increased travel time to the nearest health center increased the likelihood of children missing the SIA dose. With children living less than 15 min away from the health center as the reference group, children who lived 30 to 60 min away from a health center were 51% more likely to miss the SIA dose (PR: 1.51; 95% CI: 1.23, 1.84), while those who lived more than 1 h from the health center were 70% more likely (PR: 1.70, 95% CI: 1.37, 2.11) to miss the MR vaccination during SIA (Table 2).

## 4. Discussion

Prior to the November 2020 SIA in Zambia, 84.9% of children aged 9 to 59 months had received at least one dose of MCV. However, during the 2020 MR-SIA, only 68.6% of children in this age group received a campaign MR vaccine dose. Among the 709 measles zero-dose children, only 27.8% were vaccinated during the MR-SIA. Overall, the MR-SIA increased MCV1 coverage from 84.9% to 89.1%. An additional 6.3% of enrolled children in the PCCS received MCV2, and 58.1% received a third dose. Living in an urban area and longer travel time to a health center were associated with missing the MR-SIA vaccine dose. Zero-dose children and children whose mothers had lower education levels were less likely to be vaccinated during the SIA.

Previous MR-SIAs in Zambia reported coverages higher than 95% [17]. Although the 2020 MR-SIA used strategies similar to previous SIAs, coverage was lower for two likely reasons. First, the 2020 MR-SIA was implemented at the time when all COVID-19 control measures were instituted, including stopping all public gatherings, and when the COVID-19 vaccine was just beginning to be rolled out. During this period, a decline and delay in vaccinations were observed in many parts of the world [18]. Secondly, a Child Health Week (CHW) that included MR vaccination was conducted 5 months prior to the MR-SIA [9]. Many children may have received the MR vaccine during the CHW and therefore were not brought to the MR-SIA. Because of the general decline in the coverage of the SIA as compared to previous SIAs, the Ministry of Health must intensify routine immunization activities such as school vaccination activities and periodic intensified routine immunizations to reach children who were missed by the SIA.

According to our analysis, 27.8% of measles zero-dose children were vaccinated during the MR-SIA. This number is higher than was estimated in 11 other countries on the African continent based on an analysis of Demographic and Health Survey (DHS) data. Specifically, Portnoy et al. estimated that the proportion of measles zero-dose children vaccinated by a measles SIA ranged from 3% in Lesotho to 20% in Burkina Faso in 2003 [19]. Although PCCS and DHS do not have the same sampling frames, given that both datasets are meant to be nationally representative, this comparison may be possible and shows that Zambia’s MR-SIA does comparably well in reaching measles zero-dose children. However, further work is needed for MR-SIAs to reach more measles zero-dose children and address the inequity of routine vaccination. Ideally, SIAs do not perpetuate the inequities in routine vaccination such that SIA doses are independent of routine administered doses. This means that an SIA dose is equally as likely to reach measles zero-dose children as measles-vaccinated children; in this analysis, we expected 84.9% MR-SIA coverage among all children, regardless of previous vaccination status. An even more ambitious goal is that the SIA effectively reverse inequities in routine vaccination such that SIAs reach a larger proportion of unvaccinated children than that of all targeted children; in this analysis, we expected the proportion of measles zero-dose children reached by the MR-SIA to be greater than 84.9%. When planning SIAs, the Ministry of Health must consider utilizing and triangulating administrative coverage and local survey data to identify missed communities and develop SIA microplans that target such communities with intensive vaccination activities.

Although the MR-SIA did not vaccinate a large proportion of measles zero-dose children, it did particularly well in reaching under-immunized children who were eligible for MCV2 but had only received MCV1 at the time of the SIA. Achieving high MR-SIA coverage in the under-immunized subpopulation is a good result for the Zambia expanded program on immunization because routine coverage for MCV2 has lagged (66%) compared to routine MCV1, which is normally above 90% [4]. We postulate that the large difference in coverage between zero-dose and under-immunized subpopulations was observed because these subpopulations have different vulnerabilities and therefore require different approaches to be reached. When developing SIAs, these subpopulations must be considered carefully, and microplans must be developed according to their needs. Approximately 58% of children vaccinated during the SIA already had two MCV doses before the SIA, indicating unnecessary doses, resulting in excess costs, given that these children were fully vaccinated against measles prior to the SIA [20]. However, evidence of vaccination was based on parental recall for a considerable proportion of children; therefore, reaching these children may have provided a missing dose to children who were misclassified as vaccinated. We recommend that the vaccination card be updated to explicitly provide a section for SIA doses and dates when the SIA doses are administered.

The regression analysis showed that zero-dose children were less likely to be reached by the SIA than fully vaccinated children. In most cases, PCCSs do not report coverage among zero-dose and under-immunized children. Children who lived further from a health center or lived in an urban area were less likely to receive the MR-SIA dose. Previous studies reported increasing distance and travel time as impediments to routine vaccination [21]. Long distance to vaccination sites also affects coverage of SIAs, especially among missed communities. Our analysis confirms findings from a study conducted during the 2020 Zambia MR-SIA, which demonstrated a lower vaccination coverage among measles zero-dose children as the distance to the nearest vaccination site increased [22]. Differences in vaccination coverage between urban and rural areas have been reported previously [23]. Most countries have higher coverage in urban areas because of easy access to primary health facilities, as well as less travel time and cost [23]. Although routine MCV administrative coverage in Zambia is generally higher in urban areas than in rural areas, poor urban communities in high-population-density areas may have large numbers of unvaccinated children [24]. Not reaching zero-dose children in high-population-density urban areas may result in clusters of susceptible children that can sustain measles outbreaks. Other studies have shown that adjusting for other social-determinant factors diluted the urban–rural differential, suggesting it is not the driving factor but correlated with more proximal determinants [23]. Routine and SIA planning must deliberately address factors driving rural–urban vaccination inequalities to reach missed communities.

This study has several limitations. Inferences about the impact of the 2020 MR-SIA on reaching measles zero-dose and under-immunized children may have been constrained by recall bias. Firstly, the time between the SIA and PCCS was longer than the recommended 3 months [10]. This may have resulted in misclassification of vaccination status because caregivers may have forgotten the details of events that happened 10 months ago. Secondly, although card retention was high, SIA doses were rarely documented and could only be measured by caregivers’ recall. To address the potential impact of recall bias, we performed sensitivity analyses by restricting the analysis to only children with vaccination cards (Appendix A).

## 5. Conclusions

We have shown the MR-SIA’s strengths and limitations in addressing inequalities in immunization through reaching and failing to reach measles zero-dose and under-immunized children. The SIA reached more under-immunized children in a context in which MCV2 coverage is low. However, the SIA only reached a small proportion of measles zero-dose children. To reduce vaccination inequalities for missed communities, further work is needed for both MR-SIA and routine immunization programs in Zambia to reach more measles zero-dose children and address the inequity of routine vaccination. When developing SIAs, the vulnerability of zero-dose and under-immunized subpopulations must be considered carefully, and microplans must be developed according to their needs. One possible solution is to transition from nationwide non-selective SIAs to more targeted and selective strategies.

## Figures and Tables

**Figure 1 vaccines-11-00608-f001:**
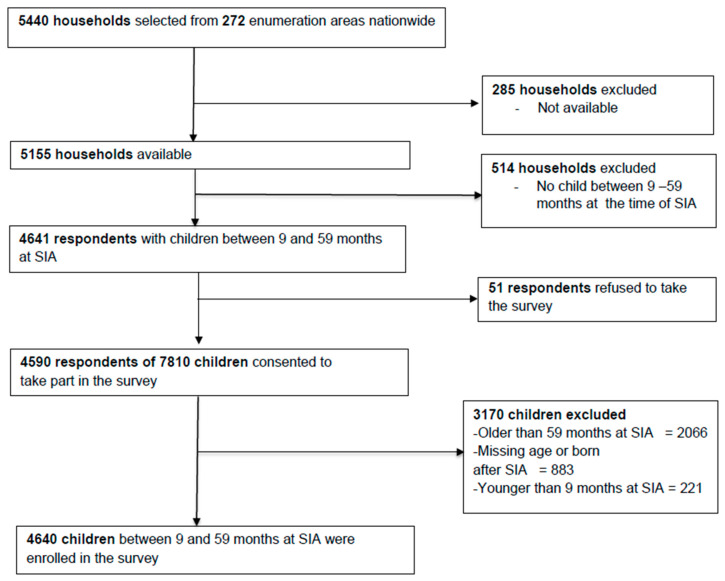
Study flow diagram.

**Figure 2 vaccines-11-00608-f002:**
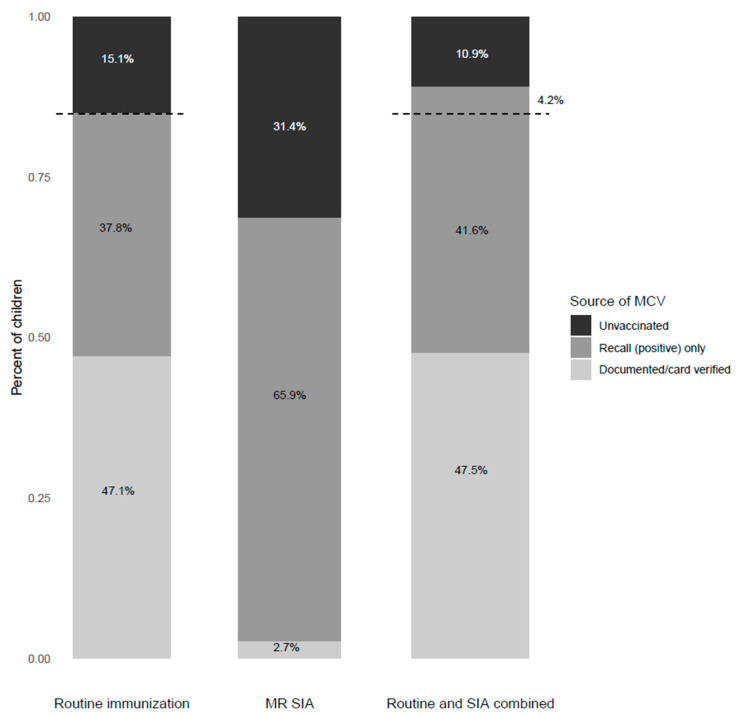
Measles and rubella vaccination coverage (at least one dose) among children between the ages of 9 and 59 months. The dashed line and percentage indicate the change in percentage of unvaccinated children before and after the MR-SIA.

**Figure 3 vaccines-11-00608-f003:**
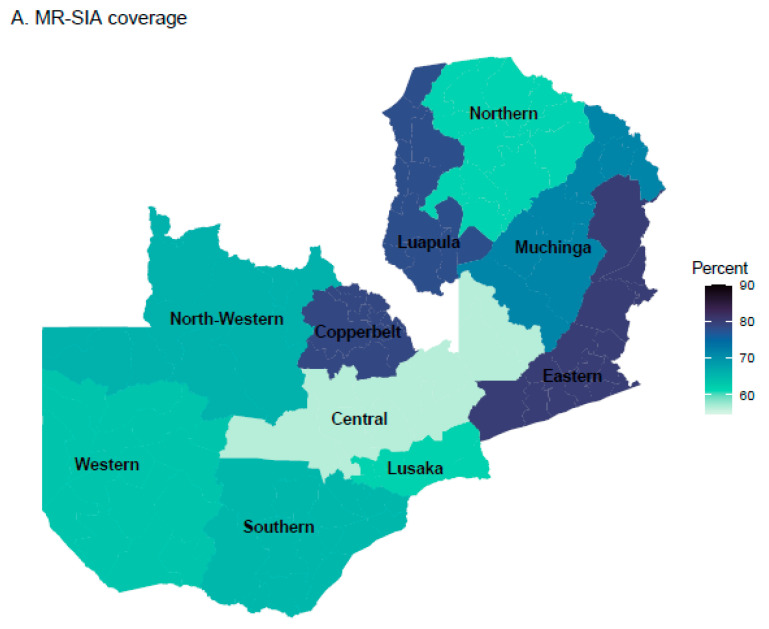
Provincial-level MR-SIA coverage and the proportion of measles zero-dose children.

**Figure 4 vaccines-11-00608-f004:**
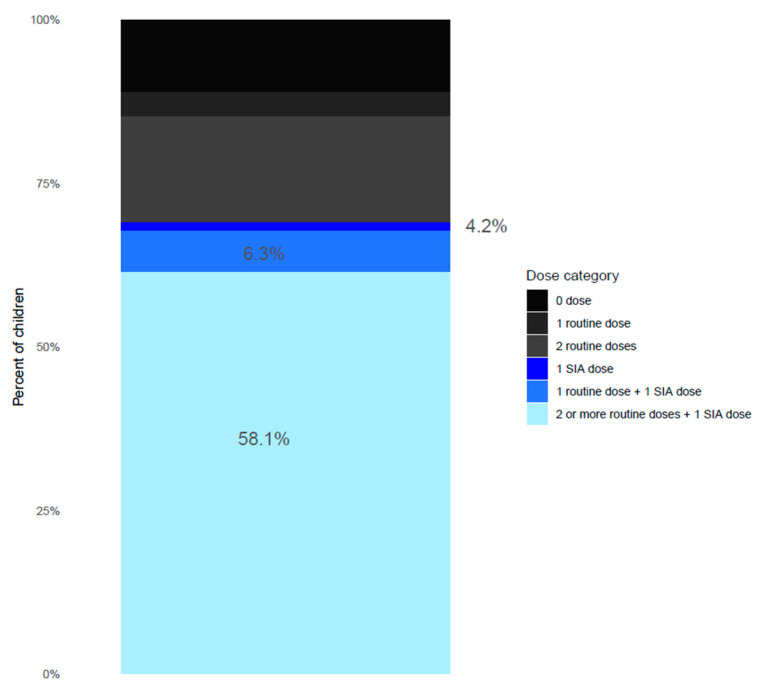
Receipt of measles and rubella vaccine among children 9–59 months post SIA, including documented or recalled evidence of vaccination. 0 dose: children who had a documented record or recalled not having received any MCV doses though routine immunization or SIA; 1 routine dose: children who had a documented record or recalled having received MCV1 through routine immunization; 2 routine doses: children who had a documented record or recalled having received MCV1 and MCV2 through routine immunization; 1 SIA dose: children who had a documented record or recalled not having received any MCV through routine immunization and a had documented record or recalled having received MCV1 through SIA; 1 routine dose + 1 SIA dose: children who had a documented record or recalled having received MCV1 through routine immunization and had a documented record or recalled having received MCV2 through SIA; 2 or more routine doses + 1 SIA dose: children who had a documented record or recalled having received MCV1 and MCV2 and/or additional MCVs through routine immunization and recalled having received an additional MCV through SIA.

**Table 1 vaccines-11-00608-t001:** Participants and household characteristics.

Characteristics	Unweighted N (%)	Weighted % (95% CI)
Total	4640	
Gender		
Male	2310 (49.8)	49.8 (48.2, 51.3)
Female	2330 (50.2)	50.2 (48.7, 51.8)
Setting		
Rural	2876 (62.0)	59.2 (57.6, 60.7)
Urban	1764 (38.0)	40.8 (39.3, 42.4)
Province		
Central	447 (9.6)	10.3 (9.4, 11.3)
Copperbelt	631 (13.6)	14.1 (13.0, 15.2)
Eastern	601 (13.0)	11.8 (10.7, 12.8)
Luapula	379 (8.2)	7.7 (6.9, 8.6)
Lusaka	816 (17.6)	17.7 (16.6, 18.9)
Muchinga	281 (6.1)	6.1 (5.4, 6.9)
Northern	387 (8.3)	8.8 (8.0, 9.7)
North-Western	292 (6.3)	5.6 (4.9, 6.3)
Southern	510 (11.0)	12.0 (10.9, 13.1)
Western	296 (6.4)	6.0 (5.4, 6.7)
Age at SIA		
9–17 months	710 (15.3)	15.4 (14.3, 16.6)
18–59 months	3930 (84.7)	84.6 (83.4, 85.7)
Availability of vaccination card		
Card available and verified	2626 (56.6)	56.6 (55.1, 58.2)
Card not available but another document verified with vaccination status	136 (2.9)	2.9 (2.4, 3.4)
Reported card available but could not provide a document with vaccination status	405 (8.8)	8.8 (7.6, 10.0)
No card and no other document	1473 (31.7)	31.7 (30.0, 33.1)
DTP vaccination status (both card and recall) ^1^		
Fully vaccinated for DTP	4076 (87.8)	88.2 (87.2, 89.2)
Under-immunized for DTP	45 (1.0)	1.0 (0.7, 1.4)
Zero-dose for DTP	97 (2.1)	2.2 (1.7, 2.6)
Unknown	424 (9.1)	8.7 (7.9, 9.6)
Head of the household		
Biological parent	3435 (74.0)	74.3 (73.0, 75.6)
Other	1205 (26.0)	25.7 (24.4, 27.1)
Maternal education level		
No education	471 (10.2)	9.6 (8.7, 10.5)
Primary	1959 (42.2)	41.0 (39.5, 42.6)
Secondary	1373 (29.6)	30.3 (28.9, 31.8)
Higher	312 (6.7)	7.7 (6.8, 8.6)
Unknown	525 (11.3)	11.4 (10.4, 12.4)
COVID-19 vaccination status of the adults in the household ^2^ (N = 9780)		
Fully vaccinated	1610 (16.5)	17.0 (16.1, 17.8)
Partially vaccinated	354 (3.6)	4.0 (3.5, 4.4)
Not vaccinated	7816 (79.9)	79.1 (78.2, 80.0)
Time to health facility for routine vaccinations		
Less than 15 min	1115 (24.0)	24.9 (23.5,26.3)
15–30 min	1188 (25.6)	25.7 (24.3, 27.0)
31 min–1 h	1204 (26.0)	26.0 (24.6, 27.4)
More than 1 h	1040 (22.4)	21.6 (20.4, 23.0)
Unknown	93 (2.0)	1.2 (0.9, 1.6)

^1^ Gavi zero-dose children are those who have not received any routine vaccines. For operational purposes, Gavi defined “zero-dose children” as children who have not received the first dose of a diphtheria–tetanus–pertussis-containing vaccine (DPT1) as a proxy measure. “Under-immunized” was defined as children missing a third dose of a diphtheria–tetanus–pertussis-containing vaccine (DPT3). ^2^ The COVID-19 vaccination status of all the adults living in the same household as the enrolled child. Fully vaccinated: one dose of Johnson & Johnson or two doses of AstraZeneca vaccine.

**Table 2 vaccines-11-00608-t002:** Risk factors associated with self-reported missing of campaign MR dose.

	Children Included in the Survey, N (%)	Children Who Missed the Campaign Measles Dose, n (%)	Unadjusted Univariable Log-Binomial Regression			Age-Adjusted Univariable Log-Binomial Regression			Multivariable Log-Binomial Regression		
Characteristic			PR ^1^	95% CI ^2^	*p*-Value	PR	95% CI	*p*-Value	PR	95% CI	*p*-Value
Total	4640	1454									
Age					0.83			—			0.5
Age 9–17 months	710 (15.3)	230 (15.8)	—	—		—	—		—	—	
Age 18–59 months	3930 (84.7)	1224 (84.2)	0.98	0.81, 1.18		—	—		0.93	0.77, 1.13	
Sex					0.45			0.4			0.4
Male	2330 (50.2)	723 (49.7)	—	—		—	—		—	—	
Female	2310 (49.8)	731 (50.3)	1.05	0.92, 1.21		1.05	0.92, 1.21		1.06	0.92, 1.22	
Setting					0.36			0.4			<0.001
Rural	2876 (62.0)	883 (60.7)	—	—		—	—		—	—	
Urban	1764 (38.0)	571 (39.3)	1.07	0.93, 1.22		1.07	0.93, 1.22		1.39	1.18, 1.63	
DTP vaccination status					<0.001			<0.001			<0.001
Fully vaccinated	4076 (87.8)	1212 (83.4)	—	—		—	—		—	—	
Under-immunized	45 (1.0)	22 (1.5)	2.31	1.24, 4.29		2.30	1.24, 4.28		2.22	1.21, 4.07	
Zero-dose	97 (2.1)	59 (4.2)	3.79	2.45, 5.87		3.80	2.45, 5.88		2.81	1.80, 4.41	
Unknown	424 (9.1)	161 (10.9)	1.61	1.27, 2.04		1.61	1.27, 2.04		1.44	1.12, 1.84	
Maternal education level					<0.001			<0.001			<0.001
No formal education	471 (10.2)	195 (13.4)	—	—		—	—		—	—	
Primary education	1959 (42.2)	604 (41.6)	0.73	0.58, 0.92		0.73	0.58, 0.92		0.73	0.58, 0.92	
Secondary education	1373 (29.6)	387 (26.6)	0.61	0.48, 0.78		0.61	0.48, 0.78		0.60	0.47, 0.77	
Higher education	312 (6.7)	92 (6.3)	0.67	0.48, 0.94		0.67	0.48, 0.93		0.66	0.46, 0.95	
Unknown	525 (11.3)	176 (12.1)	0.97	0.73, 1.29		0.97	0.73, 1.29		0.91	0.68, 1.22	
Time to health facility for routine vaccinations					<0.001			<0.001			<0.001
<15 min	1115 (24.0)	291 (20.0)	—	—		—	—		—	—	
15–30 min	1188 (25.6)	339 (23.3)	1.15	0.94, 1.40		1.15	0.94, 1.40		1.17	0.96, 1.43	
31 min to 1 h	1204 (26.0)	411 (28.3)	1.44	1.19, 1.75		1.44	1.19, 1.75		1.51	1.23, 1.84	
>1 h	1040 (22.4)	381 (26.2)	1.61	1.32, 1.97		1.61	1.32, 1.97		1.70	1.37, 2.11	
Unknown	93 (2.0)	32 (2.2)	2.17	0.60, 7.88		2.15	0.59, 7.80		1.64	0.44, 6.06	
Mother’s COVID vaccination status					0.028			0.027			
Not vaccinated	3265 (70.4)	1015 (69.8)	—	—		—	—				
Partially vaccinated	127 (2.7)	37 (2.6)	1.08	0.70, 1.65		1.08	0.70, 1.65				
Fully vaccinated	689 (14.9)	198 (13.6)	0.94	0.77, 1.14		0.94	0.77, 1.14				
Unknown	559 (12.0)	204 (14.0)	1.61	0.77, 3.40		1.60	0.76, 3.36				
Head of the household					0.072			0.072			
Biological parent	3435 (74.0)	1051 (72.3)	—	—		—	—				
Other	1205 (26.0)	403 (27.8)	1.15	0.99, 1.34		1.15	0.99, 1.34				

^1^ PR: prevalence ratio. ^2^ 95% CI: 95% confidence interval.

## Data Availability

The study protocol and dataset can be made available upon request to the corresponding author (smutemb1@jhmi.edu). Data were obtained from Zambia Ministry of Health and the Zambia National Health Research Authority under data-sharing agreements and can only be shared with permission from the Ministry of Health.

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
