# Peer review of "Challenges Addressing Inequalities in Measles Vaccine Coverage in Zambia through a Measles–Rubella Supplementary Immunization Activity during the COVID-19 Pandemic"

_vaccines, 2023, doi:10.3390/vaccines11030608_

Round 1

Reviewer 1 Report

The manuscript with manuscript ID: vaccines- 2252053 entitled "Challenges Addressing Inequalities in Measles Vaccine Cover age in Zambia through a Measles-Rubella Supplementary Immunization Activity During the COVID-19 Pandemic " is seems good research. The manuscript is well written and has sufficient data and may be published in the Vaccines after minor revision. I have many points for the authors to consider: 

·         However, this paper is self-explanatory and can be acceptable with revision after typographical and grammatical corrections.

·         The manuscript is on the whole well written but there are some problems with the English (including tenses, plurals, matching of adjectives and nouns, adverbs and verbs) such that other sections are almost ‘good’ and I would be inclined to rectify the partial contribution for plagiarism if possible.

Introduction

·         Authors indicated that line no 50-51 “Both viruses are endemic in sub-Saharan Africa and no country in the African region has eliminated measles or rubella.” As statements are confirmatory, kindly recheck and supports with more references.

Material and methods

·         What was the reason to select the time frame for PCCS that was conducted in October 2021. For the better analysis, if possible, the authors are advised to include PCCS of the year 2022 because it will provide the better information up to all COVID-19 waves. If not Please justify.

Results & Discussion:

 It is stated that Overall, 68.6% (95%CI:66.7%, 70.6%) of children eligible for the 2020 MR-SIA received MCV during the SIA. Only 2.7% of these eligible children had documented evidence of receiving the MR-SIA dose and the rest were identified by caregiver recall.”. What is the main reason for not having the vaccination document.

However, I think that all the data of MR-SIA was collected by the consent of caregiver if possible, please took the confirmation from the beneficiaries.

·          I also feel that authors should also try to focus on the Government strategy to boost up the vaccination to eliminate the inequality.

·         If the authors make the suggested changes, then accept this article because being as important contribution on Challenges Addressing Inequalities in Measles Vaccine Coverage.

Author Response

February 10, 2023

Editor

Vaccine

Dear Editor,

We are grateful for your consideration and quick review of our research article -ID vaccines- 2252053 entitled, “Challenges Addressing Inequalities in Measles Vaccine Coverage in Zambia through a Measles-Rubella Supplementary Immunization Activity During the COVID-19 Pandemic”. We have responded to all the reviewer comments. Please refer to our point by point responses to each of the reviewer comments. Further we have reviewed the entire manuscript and addressed any typographical and grammatic mistakes to improve clarity for all readers. We have also moved the tables to the appropriate sections within the manuscript submission. All the authors agree to the content of this manuscript. The authors of this manuscript have no conflicts of interest to declare.

We have uploaded the following in response to the reviewer comments:

  1. Cover letter with and response to each of the reviewer comments
  2. Track changes of the updated manuscript
  3. Clean version of the updated manuscript

Thank you for your consideration

Sincerely

Simon Mutembo

MBChB, MPH, PhD

Assistant Scientist

International Vaccine Access Center

Bloomberg School of Public Health

Johns Hopkins University

Comments and Suggestions for Authors: reviewer 1

The manuscript with manuscript ID: vaccines- 2252053 entitled "Challenges Addressing Inequalities in Measles Vaccine Cover age in Zambia through a Measles-Rubella Supplementary Immunization Activity During the COVID-19 Pandemic " is seems good research. The manuscript is well written and has sufficient data and may be published in the Vaccines after minor revision. I have many points for the authors to consider: 

  • However, this paper is self-explanatory and can be acceptable with revision after typographical and grammatical corrections.
  • The manuscript is on the whole well written but there are some problems with the English (including tenses, plurals, matching of adjectives and nouns, adverbs and verbs) such that other sections are almost ‘good’ and I would be inclined to rectify the partial contribution for plagiarism if possible.

Response 1: We reviewed the manuscript and made several revisions to the text fix any typographical and grammatical errors. We have also used the Grammarly software to proof read the manuscript

Introduction

  • Authors indicated that line no 50-51 “Both viruses are endemic in sub-Saharan Africa and no country in the African region has eliminated measles or rubella.” As statements are confirmatory, kindly recheck and supports with more references.

Response 2: We revised this sentence to “Both viruses are endemic in sub-Saharan Africa and the elimination targets for all member states in the World Health Organization (WHO) Africa region  are far from being reached” in line 50-51.

Material and methods

  • What was the reason to select the time frame for PCCS that was conducted in October 2021. For the better analysis, if possible, the authors are advised to include PCCS of the year 2022because it will provide the better information up to all COVID-19 waves. If not Please justify.

      Response 3: Zambia National Immunization Programme (NIP) has been conducting national wide non-selective MR SIAs every 4 years starting in 2003. The national wide non selective MR SIA is conducted every 4 years to avoid accumulation of a cohort of susceptible children. After every SIA, population level post-campaign coverage survey (PCCS) are conducted to estimate the population level coverage of the SIA. Last SIA was conducted in 2020 and followed by the PCCS in 2021. The next SIA and subsequent PCCS is expected in 2024. Therefore there is no 2022 SIA that we can report on.  However, an analysis beyond the scope of this manuscript has been developed using a combination of program data and mathematical models to assess the impact of COVID-19 on the SIA.

Results & Discussion:

 It is stated that Overall, 68.6% (95%CI:66.7%, 70.6%) of children eligible for the 2020 MR-SIA received MCV during the SIA. Only 2.7% of these eligible children had documented evidence of receiving the MR-SIA dose and the rest were identified by caregiver recall.”. What is the main reason for not having the vaccination document.

However, I think that all the data of MR-SIA was collected by the consent of caregiver if possible, please took the confirmation from the beneficiaries.

Response 4: We acknowledge that documentation of the SIA doses of the MR vaccine was poor and this may have resulted in recall bias during the PCCS and we highlight this as a limitation in line 122 - 123. In addition to not documenting the SIA doses, we  found that 32% children did not have their vaccination documents during the interview.  The main reason for poor documentation of the SIA doses is that the current vaccination card in Zambia only has two lines/spaces for the MCV1 and MCV2. Which means if children received their 3rd or 4th dose from the SIA, this information is less likely to be recorded on the vaccination card. In addition to this even though the vaccination card has a section for “other vaccines” it does not explicitly provide instructions to record SIA doses in that section.

We recognize that this is a weakness in the evaluation of the coverage of the SIA and follow up of children and therefore we edited the manuscript and recommend that the vaccination card must be updated to include a section for recording SIA doses (line 272 – 273).  

Caregiver refers to parent and and legal guardian. Since the beneficiaries or the receipients of the vaccines are children below the age of 5 years we could not confirm from these children.

  • I also feel that authors should also try to focus on the Government strategy to boost up the vaccination to eliminate the inequality.

Response 4: We have updated the manuscript and made specific recommendations focused on the program to address some of the vaccination inequalities. Please refer to lines 242 - 244, 258 - 260, 272 - 273, 301 - 309·      

  If the authors make the suggested changes, then accept this article because being as important contribution on Challenges Addressing Inequalities in Measles Vaccine Coverage.

Reviewer 2 Report

The problem question is what are challenges addressing inequalities in measles vaccine cover- age in Zambia through a measles-rubella supplementary immunization activity during the COVID-19 pandemic?

 The topic of the manuscript is relatively original and is within the scope of the Journal and could be valuable to the scientific audience. Therefore, is important to understand how effective is a measles-rubella supplementary immunisation activity (MR-SIA) , which is conducted to address inequalities in coverage and fill population immunity gaps when routine immunisation services fail to reach all children with two doses of measles-containing vaccine (MCV) in Zambia.

 This study bridging the gap because there were examined inequalities in measles vaccine cover-age in Zambia through a measles-rubella supplementary immunization activity during the COVID-19 pandemic.

 The title of the article is accurate. Abstract reflects the work done and the conclusions drawn. Conclusions are consistent with the evidence and arguments presented. The references are appropriate. Some clarifications are however needed:

 1. Research hypothesis is missing. Authors should provide justification for the research hypothesis.

 2. It is not clear why there is a reference to Table 4 that is not presented in the article.

 3. I suppose that not only the limitations of the study could be defined but also future prospects should be described.

 4. My suggestion Abbreviations (e.g. PR, CI) must be explained below the Table 2.

 TO SUM UP I think the author(s) need to make the recommended corrections.

Author Response

March 3, 2023

Editor

Vaccine

Dear Editor,

We are grateful for your consideration and quick review of our research article -ID vaccines- 2252053 entitled, “Challenges Addressing Inequalities in Measles Vaccine Coverage in Zambia through a Measles-Rubella Supplementary Immunization Activity During the COVID-19 Pandemic”. We have responded to all the reviewer comments. Please refer to our point by point responses to each of the reviewer comments. Further we have reviewed the entire manuscript and addressed any typographical and grammatic mistakes to improve clarity for all readers. We have also moved the tables to the appropriate sections within the manuscript submission. All the authors agree to the content of this manuscript. The authors of this manuscript have no conflicts of interest to declare.

We have uploaded the following in response to the reviewer comments:

  1. Cover letter with and response to each of the reviewer comments
  2. Track changes of the updated manuscript
  3. Clean version of the updated manuscript

Thank you for your consideration

Sincerely

Simon Mutembo

MBChB, MPH, PhD

Assistant Scientist

International Vaccine Access Center

Bloomberg School of Public Health

Johns Hopkins University

Comments and Suggestions for Authors: reviewer 2

The problem question is what are challenges addressing inequalities in measles vaccine coverage in Zambia through a measles-rubella supplementary immunization activity during the COVID-19 pandemic.

The topic of the manuscript is relatively original and is within the scope of the Journal and could be valuable to the scientific audience. Therefore, is important to understand how effective is a measles-rubella supplementary immunisation activity (MR-SIA) , which is conducted to address inequalities in coverage and fill population immunity gaps when routine immunisation services fail to reach all children with two doses of measles-containing vaccine (MCV) in Zambia.

 This study bridging the gap because there were examined inequalities in measles vaccine coverage in Zambia through a measles-rubella supplementary immunization activity during the COVID-19 pandemic.

 The title of the article is accurate. Abstract reflects the work done and the conclusions drawn. Conclusions are consistent with the evidence and arguments presented. The references are appropriate. Some clarifications are however needed:

  1. Research hypothesis is missing. Authors should provide justification for the research hypothesis.

Response 1: We have edited the introductory section and stated the aim of the analysis more explicitiy and provide a justification we added the study goal and hypothesis to the last paragraph of the introduction section (line 71-76) of the manuscript below:

“We used  national, cross-sectional data from the PCCS to understand how MR-SIA can address vaccination inequalities by: estimating routine and SIA MCV coverages;  estimating the proportion of measles zero-dose and under-immunized children reached by the MR-SIA, and identified reasons associated with missing the MR-SIA.  Our study will help program managers and researchers understand vaccination inequalties that are overlooked when implementing SIA and when measuring the impact of the SIAs using the PCCS.”

  1. It is not clear why there is a reference to Table 4 that is not presented in the article.

Response 2: We should have reference Table 2 not Table 4 in the result section. We have made amendments in the manuscript and resubmit the revised manuscript.

  1. I suppose that not only the limitations of the study could be defined but also future prospects should be described.

Response 3: We have tried to address the issues described above as well as those raised by other reviewers in the discussion. We have added several sentences in the conclusion section to summarized what we think are influential for recommendations and future prospects. Please refer to Lines . Please refer to lines 242 - 244, 258 - 260, 272 - 273, 301 - 309,

  1. My suggestion Abbreviations (e.g. PR, CI) must be explained below the Table 2.

Response 4: This has been addressed in footnotes for table 2

 TO SUM UP I think the author(s) need to make the recommended corrections.